# An apicobasal gradient of Rac activity determines protrusion form and position

Africa Couto[1],*, Natalie Ann Mack[1],*, Lucrezia Favia[1] & Marios Georgiou[1]

Each cell within a polarized epithelial sheet must align and correctly position a wide range of subcellular structures, including actin-based dynamic protrusions. Using *in vivo* inducible transgenes that can sense or modify Rac activity, we demonstrate an apicobasal gradient of Rac activity that is required to correctly form and position distinct classes of dynamic protrusion along the apicobasal axis of the cell. We show that we can modify the Rac activity gradient in genetic mutants for specific polarity proteins, with consequent changes in protrusion form and position and additionally show, using photoactivatable Rac transgenes, that it is the level of Rac activity that determines protrusion form. Thus, we demonstrate a mechanism by which polarity proteins can spatially regulate Rac activity and the actin cytoskeleton to ensure correct epithelial cell shape and prevent epithelial-to-mesenchymal transitions.

[1] School of Life Sciences, University of Nottingham, Nottingham NG7 2UH, UK. * These authors contributed equally to this work. Correspondence and requests for materials should be addressed to M.G. (email: marios.georgiou@nottingham.ac.uk).

Epithelial sheets exhibit several defining characteristics that enable their correct function. These include mechanically strong cell–cell junctions that provide adhesive links between cells and ensure epithelial strength and integrity; and a coordinated cell polarity, which imparts correct cell shape and tissue organization. These characteristics allow epithelia to serve as effective barriers whilst also maintaining plasticity, which is essential to accommodate changes in tissue organization, required both during homeostasis and during major morphogenetic movements, such as cell intercalation or epithelial bending[1]. Key to the acquisition of these characteristics is the intimate interplay between adhesion (both integrin- and cadherin-mediated[2]), polarity proteins and regulators of the actin cytoskeleton, thereby allowing each cell within the sheet to align their apical–basal axes and to correctly position a wide range of subcellular structures and activities across the entire tissue. These include the correct positioning of cell–cell junctions and of distinct cortical membrane compartments[3,4], as well as of actin-based dynamic protrusions[5].

Rho family GTPases are known to control the formation of a variety of actin filament-based structures[6] and it has been shown in many systems that apically localized polarity proteins, Rho GTPases and cell–cell junctions act in concert to correctly regulate cell polarity and cytoskeletal organization[7]. This has been shown very effectively when using the *Drosophila* pupal notum as a model system to study a three-dimensional polarized epithelium in the living animal[5,8–10].

By combining genetic and cell biological analyses we have previously shown that epithelial cells within the fly notum possess distinct classes of actin-rich dynamic protrusion along their apical–basal axis. Cells possess apical microvillar-like protrusions, lateral sheet-like protrusions at an intermediate level, and filopodia and lamellipodia at the base of the cell[5]. We found that apical polarity proteins are required to cooperate with Rho GTPases to control cell morphology and to form and position these distinct classes of dynamic protrusion[5]. Cdc42–Par6–aPKC and Bazooka/Par3 (Baz) appear to have antagonistic roles in the formation of basolateral protrusions. Cdc42–Par6–aPKC is required for actin filament formation and protrusion dynamics, whereas Baz functions to inhibit actin polymerization, via inhibition of the Rac-GEF Sif/TIAM1 (ref. 5). This Baz-mediated inhibition of Rac activity, via TIAM1, has been shown in several systems, to regulate protrusions in mammalian fibroblasts and neuronal cells[11,12] and during tight junction assembly in polarizing MDCK cells[13,14].

Recent studies have demonstrated the importance of a spatiotemporal regulation of Rho GTPase signalling for correct apicobasal polarization. In polarizing MDCK cells, higher levels of Rac activity have been observed at the lateral membrane when compared with the apical[15] and at adherens junctions, when compared with the more apical tight junctions[14]. A similar differential regulation of Rac activity was also observed in intestinal epithelial cells[16]. Therefore, recent work seems to imply that Rac activity is spatially tightly regulated in epithelial cells and that Baz could be a key player in mediating this regulation. In this study, we imaged the epithelium of the developing *Drosophila* pupal notum, expressing *in vivo* inducible constructs that can sense or modify Rac activity, to demonstrate an apicobasal gradient of Rac activity that is required to correctly form and position distinct classes of protrusion along the apicobasal axis of the cell. Apicobasal polarity is required to form this gradient, and we show that we can modify the Rac activity gradient in genetic mutants for specific polarity proteins, with consequent changes in protrusion form and position. We additionally show, using photoactivatable Rac transgenes, that it is the level of Rac activity that determines protrusion form, with high levels of Rac activity

required to form filopodia and lower levels required to form lamellipodia. It has long been known that polarity proteins are essential in maintaining epithelial cell shape, with loss of apicobasal polarity being a prerequisite for epithelial-to-mesenchymal transitions and abnormal cell invasiveness[17]. Our data provide a mechanism by which polarity proteins can influence Rac activity and the actin cytoskeleton, thereby ensuring the maintenance of epithelial cell shape, and consequently the architecture of the epithelium as a whole.

## Results

**An apicobasal gradient of Rac activity.** Our model system allows us to study epithelial cell shape in the living animal in extremely high spatial and temporal resolution. To do this, we combine powerful *Drosophila* genetic techniques with confocal imaging of a living fly at pupal stages. Specifically, we combine the Flp/FRT system[18], the MARCM technique[19] and Neuralized-Gal4 (ref. 20) to express GFP-fusion proteins (or other constructs) in well-spaced epithelial cells in the fly notum (Fig. 1a–a″). For example, using the Neuralized-Gal4 driver together with the UAS-GFP:Moe reporter construct (consisting of the actin binding domain of Moesin fused to GFP) we were able to label actin filaments in these isolated epithelial cells. Neuralized-Gal4 drives expression specifically within the precursor cells for the external sensory organ. Importantly, at the time of our analysis (12–15 h after puparium formation (APF)), these epithelial cells are not yet fully committed to a precursor fate[9]. Thus, these well-spaced GFP-positive cells are representative of all cells within the epithelium. This assertion is supported by the fact that epithelial cell shape, protrusion form and position, and protrusion dynamics are identical irrespective of whether cells are Neuralized-positive or -negative (Fig. 1b–b″ (ref. 5)).

Using this system we observed distinct classes of actin-rich dynamic protrusions along the apicobasal axis of these cells: apical microvillar-like protrusions, lateral intermediate-level protrusions, observed ∼3–6 μm below the cell apex, and basal filopodia and lamellipodia (Fig. 1a). The lateral intermediate-level protrusions lie beneath the adherens junctions, in the basolateral domain and like the basal filopodia and lamellipodia, are dynamic. These intermediate level protrusions resemble arm-like (Fig. 1c) or sheet-like (Fig. 1d) lateral protrusions when imaged in the x–z plane, that extend between the surrounding cells, and have a highly distinct morphology from the very thin filopodia observed at the base of the cell.

We have previously shown that the apical polarity proteins, Par6 and aPKC, the Rho GTPases, Rac and Cdc42, and the actin regulator complexes SCAR/WAVE and Arp2/3 are all required to form dynamic actin-rich protrusions in the basolateral domain of the cell, but are not required for the actin enrichment observed at the cell apex[5]. We therefore observe a graded distribution of protrusions, with no Rac-induced protrusions present apically, sheet-like protrusions predominating at an intermediate level and filopodia restricted to the base of the cell. We have also shown that in cells with elevated levels of Rac activity (for example, in cells expressing Rac$^{V12}$, or in *baz* mutants) filopodia are no longer restricted to the base of the cell, and are observed throughout the basolateral domain, occurring at the expense of the intermediate level sheet-like protrusions[5]. These results indicate that the level of Rac activity has a profound effect on protrusion form.

Since Baz is an apical polarity protein that localizes to the adherens junction in flies, we posited that Baz-mediated inhibition of Rac activity could be required to generate a gradient of Rac activity along the apicobasal axis of epithelial cells (Fig. 1e). This gradient of Rac activity, with high levels of Rac at the base of the cell, could account for the observed graded distribution of

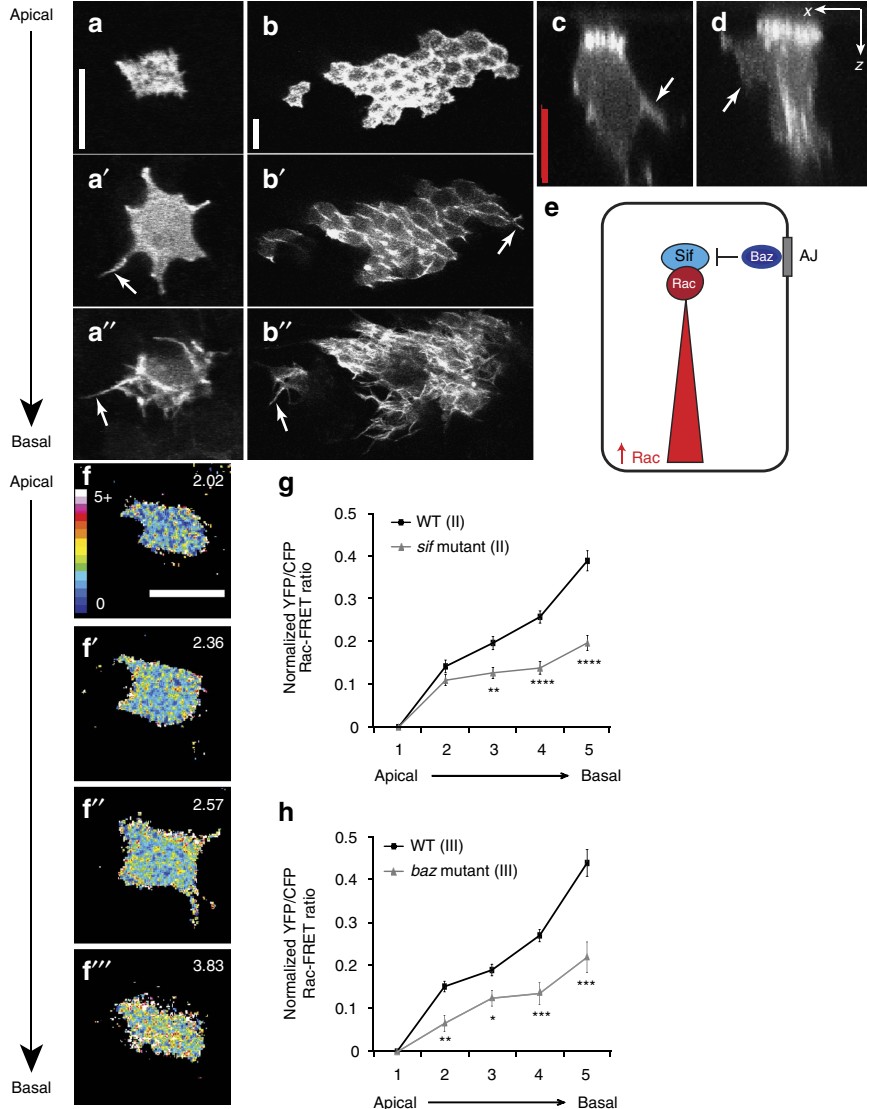

**Figure 1 | A gradient of Rac activity corresponds with a graded distribution of dynamic protrusions.** (**a**–**a″**) Live imaging of a GFP:Moe-labelled epithelial cell. GFP:Moe localizes to the actin cytoskeleton and highlights the presence of short apical microvillar-like protrusions (**a**), lateral protrusions 5 μm from the cell apex (**a′**) and basal protrusions at the bottom of the cell (10 μm from apex, **a″**). (**b**–**b″**) All epithelial cells possess apical, intermediate and basal protrusions. MARCM clones using Pannier-Gal4 generated random patches of GFP:Moe-labelled cells in the fly notum. All labelled cells, irrespective of their position on the notum, possessed all three classes of dynamic protrusion. (**c,d**) Lateral intermediate-level protrusions have an arm-like (**c**) or sheet-like morphology (**d**) when imaged in the *x–z* plane (arrows). (**e**) Schematic diagram representing our working model: Baz-mediated inhibition generates a gradient of Rac activity along the apicobasal axis of the cell. (**f**–**f‴**) Ratiometric FRET images demonstrating differential Rac activity along the apicobasal axis of a wild-type cell. The mean FRET ratio of each *z*-section is displayed (top right). (**g,h**) Quantification of Rac-FRET efficiencies demonstrates an apicobasal gradient of Rac activity in wild-type (WT) cells that is disrupted in *sif* and *bazooka* (*baz*) mutant cells. Plotted are the mean YFP/CFP Rac-FRET ratios along the apicobasal axis, normalized to the values of the apical region of the cell (where 1 is most apical and 5 most basal; see Methods for more details). (**g**) WT and *sif* mutant cells using a Rac-FRET construct inserted on the second chromosome (*n* = 27 cells from four animals for each genotype). (**h**) WT (*n* = 30 cells from ten animals) and *baz* mutant (*n* = 24 cells from 11 animals) using a Rac-FRET construct inserted on the third chromosome. All error bars represent s.e.m. Scale bars, 10 μm (white), 5 μm (red). Student's *t*-test was performed to determine statistical significance and *P* values are shown on graph. *P* > 0.5 was considered not significant, \**P* < 0.05, \*\**P* < 0.01, \*\*\**P* < 0.001, \*\*\*\**P* < 0.0001.

Rac-induced protrusions in wild-type cells. To directly test this hypothesis we took advantage of a Rac Förster resonance energy transfer (FRET) biosensor, which has previously been used *in vivo* in both the zebrafish and the fly[21,22] (Supplementary Fig. 1a). When expressed in epithelial cells of the fly notum using Neuralized-Gal4, the biosensor was distributed uniformly throughout Neuralized-positive cells, with cell protrusions clearly identifiable (Supplementary Fig. 1b). This uniform localization of the protein mirrors the subcellular localization of this probe when previously imaged in other *in vivo* systems[21,22].

This FRET biosensor is a modified version of Raichu-Rac[22,23] and is not tethered to the membrane and therefore localization within the cytosol is to be expected. However, as expected, within a single confocal section, we observed a higher FRET signal and therefore higher levels of Rac activity within protrusions and at the cell cortex (Supplementary Fig. 1c,d).

When we calculated the FRET signal in individual confocal sections across the apicobasal axis of the cell we consistently observed a strong gradient of Rac activity, with high levels of Rac activity at the base of the cell (Fig. 1f–h, Supplementary Fig. 1g,h).

Control experiments demonstrated that this gradient of Rac activity was robust and reproducible (Supplementary Fig. 1e,f). Based on our hypothesis that Baz-mediated inhibition of Sif/TIAM1 was responsible for generating this gradient of Rac activity, we would expect the gradient to be severely disrupted in *sif/tiam1* or *baz* mutant cells. We utilized Neuralized-Gal4 to express the Rac FRET biosensor specifically in well-spaced cells in *sif/tiam1* mutant animals and the MARCM technique together with Neuralized-Gal4 to express the biosensor in individual cells in *baz* mutant clones. In both cases we observed a significant flattening of the Rac activity gradient throughout the basolateral domain (Fig. 1g,h, Supplementary Fig. 1g,h).

**Polarity proteins regulate the Rac activity gradient.** It has long been established in a wide variety of systems that the maintenance of apicobasal polarity relies on the mutual exclusion of proteins that define the apical and basolateral domains of a cell[24]. Basolateral polarity proteins are known to restrict the extent of the apical domain[25]. In *lethal giant larvae* (*lgl*) or *discs large* (*dlg*) mutants apical and junctional proteins have been shown to basally mislocalise in several *Drosophila* tissues, such as in the embryo, imaginal discs, and the follicular epithelium[4,26–28]. We wanted to see whether *lgl* or *dlg* mutants affect cell polarity in cells of the epithelium of the fly pupal notum, and whether this has any effect on epithelial cell shape and/or protrusion formation or positioning. We generated clones of cells mutant for *lgl* or *dlg* and labelled well-spaced epithelial cells in the fly pupal notum using GFP:Moe. We found that in epithelial cells mutant for either *dlg* or *lgl*, intermediate level protrusions were specifically

lost (Fig. 2). In *dlg* mutant cells, actin enrichment was still observed at the cell apex (Fig. 2b) and basal level filopodia were morphologically normal and possessed wild-type protrusion dynamics, with filopodial extension/retraction rates of $2.9 \pm 0.9\,\mu m\,min^{-1}$ (Fig. 2b–b″, Supplementary Fig. 2b,d, and Supplementary Movie 1). However, at an intermediate level (3–6 µm from the cell apex) the sheet-like protrusions that are observed in wild-type cells are specifically lost (compare Fig. 2a′,b′; see Fig. 2f for protrusion distributions). When imaged in *xz*, the lateral sheet-like protrusions that are readily observable in wild-type cells (Fig. 1c,d) cannot be seen in *dlg* mutant cells (Fig. 2d). A very similar phenotype was observed in *lgl* mutant epithelial cells (Fig. 2c–c″) with again intermediate level protrusions being specifically lost (Fig. 2c,e,g). However, it was evident that in *lgl* mutant cells, basal protrusions exhibited defects in both protrusion morphology and dynamics. Basal filopodia were frequently longer and thicker than wild type and possessed very limited dynamics (Fig. 2e, Supplementary Fig. 2c,d and compare Supplementary Movies 2 and 3 for protrusion dynamics in *lgl* mutant and wild-type cells respectively). This effect on protrusion morphology and dynamics could reflect Lgl's known role in regulating the actomyosin cytoskeletal network and its known direct interaction with non-muscle myosin II-A (refs 29,30).

The specific loss of intermediate level protrusions in basolateral polarity protein mutants led us to postulate that perhaps this loss was due to a modification of the Rac activity gradient. If, as observed in other tissues, there were an expansion of the apical and/or junctional domain in *lgl* and *dlg* mutants this would lead to a more basal localization of Baz, which would lead to a more

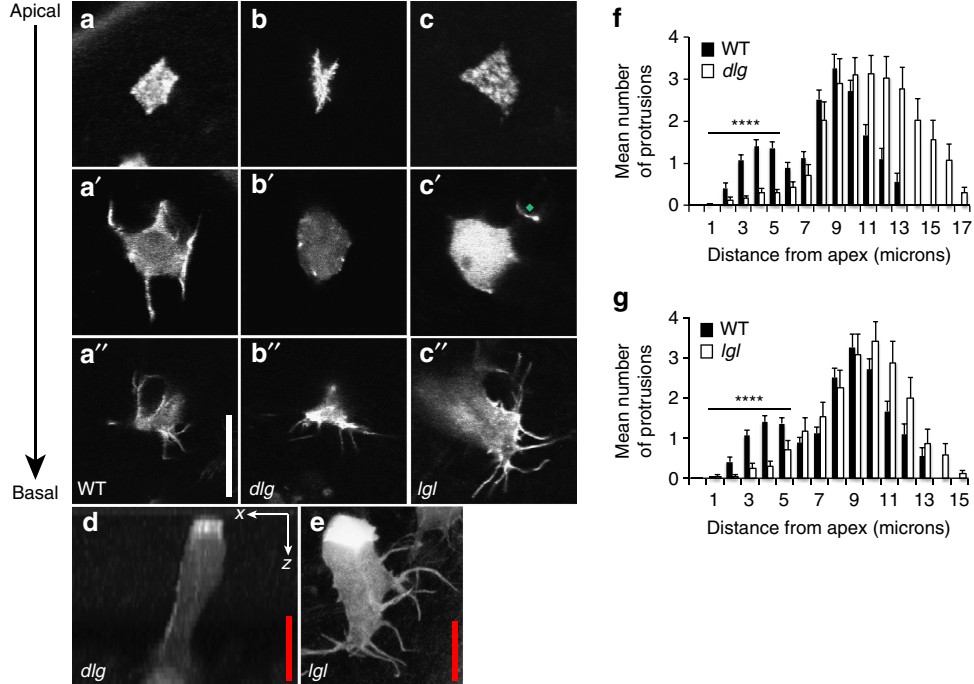

**Figure 2 | Loss of basolateral polarity proteins dlg or lgl leads to the specific loss of lateral protrusions.** (**a–a″**) Live imaging of a GFP:Moe-labelled wild-type epithelial cell showing apical (**a**) intermediate (**a′**) and basal (**a″**) confocal sections. (**b,c**) Apical, intermediate and basal confocal sections of GFP:Moe-labelled *dlg* (**b–b″**) and *lgl* (**c–c″**) mutant cells. Note the specific loss of protrusions in the intermediate sections. The green diamond highlights a protrusion that originates at a more basal plane. (**d**) An *x–z* confocal slice in a *dlg* mutant clone; (**e**) a *z*-projection of an *lgl* mutant cell, each again highlighting a lack of lateral protrusions. Note the abnormally thick and long basal protrusions in the *lgl* mutant cell (**e**). (**f,g**) Protrusion distributions for *dlg* (**f**) and *lgl* (**g**) mutant cells, highlighting a lack of intermediate level protrusions. Mutant cell distributions are represented by white bars, wild-type cell distributions by black bars. Error bars represent s.e.m. Wild-type *n* = 50 cells from ten animals; *dlg n* = 35 mutant cells from 19 animals; *lgl n* = 40 mutant cells from 17 animals. Scale bars, 10 µm (white), 5 µm (red). Student's *t*-test was performed to determine statistical significance and *P* values are shown on graph. *P* > 0.5 was considered not significant, *$P$ < 0.05, **$P$ < 0.01, ***$P$ < 0.001, ****$P$ < 0.0001.

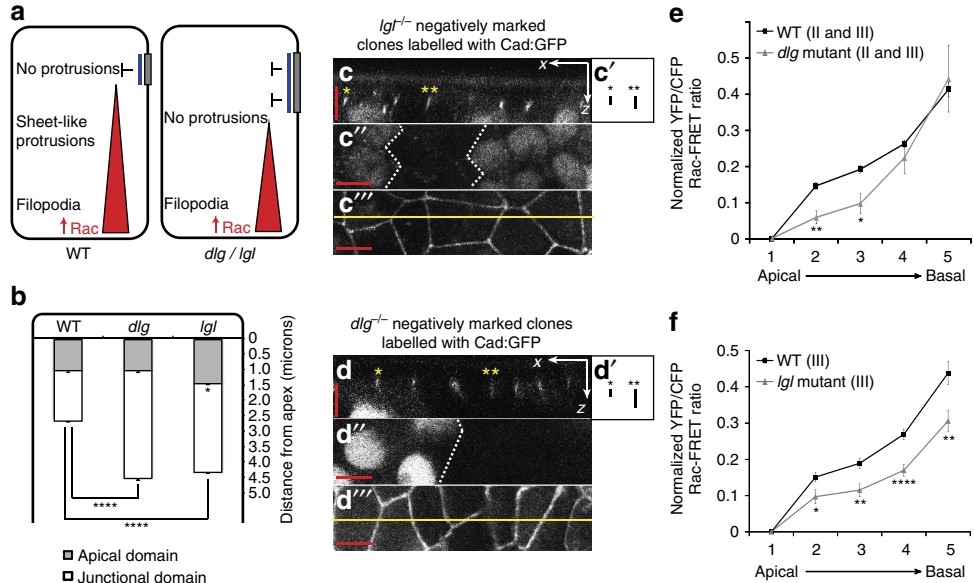

**Figure 3 | An expansion of the junctional domain affects the Rac activity gradient.** (**a**) Schematic diagram illustrating how, in *dlg* or *lgl* mutant cells, an expansion of the adherens junction could lead to a steepening of the Rac activity gradient and affect protrusion distribution. (**b**) A significant expansion of the apical and junctional domains was observed in *lgl* mutant cells, and a significant expansion of the junctional domain was observed in *dlg* mutant cells ($n = 96$ junctions from eight animals for each genotype). (**c,d**) Example *x–z* confocal slices illustrating Cad:GFP localization within *lgl* (**c**) or *dlg* (**d**) mutant clones. (**c′,d′**) Boxes highlight expansion of mutant junctions (**\*\***) in comparison to neighbouring wild-type junctions (**\***). (**c″,d″**) A lack of nuclear GFP represents location of homozygous mutant clones. (**c‴,d‴**) *x–y* confocal slice; yellow line shows region of *x–z* slice shown in **c** and **d**. (**e,f**) Quantification of Rac-FRET efficiencies. Plotted are the mean YFP/CFP Rac-FRET ratios along the apicobasal axis, normalized to the values of the apical region of the cell (where 1 is most apical and 5 most basal; see Methods for more details). (**e**) WT ($n = 57$ cells from 14 animals) and *dlg* mutant ($n = 10$ cells from eight animals) using Rac-FRET constructs inserted on either the second or third chromosomes. (**f**) WT ($n = 30$ cells from ten animals) and *lgl* mutant ($n = 30$ cells from 15 animals) using a Rac-FRET construct inserted on the third chromosome. All error bars represent s.e.m. Scale bars, 5 μm. Student's *t*-test was performed to determine statistical significance and *P* values are shown on graph. $P > 0.5$ was considered not significant, $*P < 0.05$, $**P < 0.01$, $***P < 0.001$, $****P < 0.0001$.

basal inhibition of Rac activity and a steepening of the Rac activity gradient (Fig. 3a). We first wanted to confirm that the effects we see on protrusion form and position in *lgl* and *dlg* mutants are due to a disruption of apicobasal polarity. To test this we generated small clones of tissue in the pupal notum, homozygous mutant for either *lgl* or *dlg*, and observed the effect on apicobasal polarity, in both live (using GFP reporters) and fixed tissue (using antibodies to polarity protein determinants). For live imaging, we combined two ubiquitously expressed markers conjugated with GFP:Resille-GFP, which labels the cell membrane[31], and E-Cadherin-GFP, which labels the adherens junctions. In this way we could monitor the extent of both the apical and junctional domains of the cell both inside and outside the mutant clones. Using this methodology we observed a significant expansion of the apical domain in *lgl* mutant cells and a significant expansion of the junctional domain in both *dlg* and *lgl* mutant cells (Fig. 3b–d). Importantly, we observed co-localization of Baz and E-cadherin, even in expanded junctions in mutant cells (Supplementary Fig. 3). This demonstrates that an expansion of the junction in *dlg* and *lgl* mutant cells results in an expansion of Baz localization at the cell cortex, which could lead to a corresponding expansion of Baz-mediated inhibition of Rac activity (Fig. 3a).

Using the MARCM system together with Neuralized-Gal4 we expressed the Rac FRET biosensor in *dlg* or *lgl* mutant cells in the fly notum at pupal stages. In *dlg* mutant cells we observed a significant reduction in Rac activity at intermediate levels (3–6 μm from the cell apex), however Rac activity was found to be at wild-type levels at the base of the cell (Fig. 3e, Supplementary Fig. 3c), representing a steepening of the Rac activity gradient within the basolateral domain. The Rac activity

gradient was again altered in *lgl* mutant cells, although the effect on Rac activity was markedly different from that of *dlg*. In *lgl* mutant cells we saw a significant reduction in Rac activity throughout the basolateral domain (Fig. 3f, Supplementary Fig. 3d). Although the level of Rac activity is significantly reduced at the base of the cell in *lgl* mutant cells, the profile of the Rac gradient is very different from that of *sif*, where Rac activity is more severely disrupted (Fig. 1g) and protrusions are lost throughout the basolateral domain[5]. We therefore have observed three distinct Rac activity profiles that correlate with the observed effect on protrusion form and position: a loss of intermediate protrusions with normal dynamic filopodia at the base of the cell (*dlg* mutant cells); a loss of intermediate protrusions with abnormal protrusion morphology and dynamics at the base of the cell (*lgl* mutant cells); and a loss of protrusions throughout the basolateral domain (*sif* mutant cells, compare Figs 1g and 3e,f).

**The level of Rac activity determines protrusion form.** Much of the work outlined above provides indirect evidence to suggest that it is the level of Rac activity that determines protrusion form. This would explain the graded distribution of protrusions observed along the apicobasal axis of wild-type cells (Fig. 3a). To directly address this issue, we made use of recently developed photoactivatable forms of Rac (PA-Rac)[21,32], which we over-expressed in clones of epithelial cells in the fly notum using MARCM. These tools consist of a constitutively active Rac1 (PA-RacQ61L), or a dominant negative Rac1 (PA-RacT17N) fused to the photoreactive LOV (light oxygen voltage) domain from phototrophin, sterically blocking Rac1 interactions. The LOV domain interacts with a carboxy-terminal helical extension

(Jα) in the dark, blocking Rac1 interaction with downstream effectors. Photon absorption causes dissociation and unwinding of the Jα helix, thereby releasing steric inhibition (Supplementary Fig. 4a). We also used a light insensitive version of the construct (a single amino acid substitution in the LOV domain (C450M) renders the protein light insensitive), which we used as a control. PA-Rac has been shown to allow the rapid and reversible activation or inactivation of Rac using light, both *in vitro*[32] and *in vivo* in *Drosophila*[21]. These tools have allowed us to evaluate the effect of locally activating or inhibiting Rac activity simply by exposing a small region of the cell to repeated pulses of blue light.

When overexpressing PA-Rac constructs in small clones in the fly pupal notum, we found that as natural light incorporates the correct wavelengths to activate PA-Rac, the simple act of overexpression led to some level of photoactivation, without the need to photoactivate using a confocal laser. This led to mild but interesting phenotypes in the cells that were expressing the PA-Rac constructs. We found that those cells expressing PA-RacT17N possessed a significantly greater proportion of lamellipodial protrusions at the base of the cell when compared to those cells expressing the PA-RacQ61L constitutively active construct (Supplementary Fig. 4b). We have previously observed the same effect when genetically reducing Rac activity in these cells (for example, removing a single copy of the three *Drosophila* Racs, or the Rac GEF *sif*[5]). Protrusions in PA-RacT17N expressing cells also possessed an abnormal morphology when compared to protrusions from wild-type cells or PA-RacQ61L expressing cells (Supplementary Fig. 4c,d). When expressing the photoactivatable constitutively active Rac construct we frequently observed cells that had delaminated from the epithelium. Epithelial cells do delaminate from this epithelium at early pupal stages, however this delamination is concentrated at the midline region and is rapidly followed by cell death[33]. PA-RacQ61L expressing cells were found to frequently delaminate (23% of all expressing cells delaminate), irrespective of their position within the epithelial sheet. These delaminated cells take on a front-rear polarity, are motile, and do not undergo immediate cell death (we have imaged delaminated cells for up to 45 min without observing cell death) demonstrating the importance of Rac activity levels in epithelial-to-mesenchymal transitions.

Our results have led us to propose that (i) the level of Rac activity is a key factor in determining protrusion form and (ii) the formation of sheet-like lamellipodial protrusions require lower levels of Rac activity than filopodial protrusions. If these statements are correct, by increasing Rac activity we should be able to generate lamellipodial protrusions, and by increasing Rac activity further we should be able to convert lamellipodial protrusions into filopodial protrusions. As delaminated cells were frequently observed in PA-RacQ61L expressing cells, we carried out photoactivation experiments on both delaminated cells and cells within the epithelial sheet, and observed profound effects on protrusion morphology on both types of cell. Following exposure to repeated pulses of laser light, delaminated epithelial cells lacking lamellipodial protrusions rapidly sprouted large, ruffling lamellipodia (Fig. 4a,d, Supplementary Movie 4). On observing delaminated epithelial cells that already possessed lamellipodial protrusions, we could rapidly convert these protrusions into filopodia following illumination (Fig. 4b,e, Supplementary Movie 5). When photoactivating epithelial cells that remained within the epithelium, which retained apicobasal polarity and a columnar cell shape, the effect was less pronounced. However, the same effect could be seen: photoactivation of lamellipodia led to filopodial growth with a concurrent reduction in lamellipodial area (Fig. 4c,f, Supplementary Movie 6). However, cells within the epithelium did require greater levels of photoactivation to convert lamellipodia into filopodia: delaminated cells took, on average, less than two rounds of photoactivation to reduce lamellipodial area by 50%, with a concurrent increase in average filopodial length of over 200% (Fig. 4e); while for cells that remained within the epithelial sheet, it took an average of nine rounds of photoactivation to reduce lamellipodial area to 50%, with a concurrent increase in average filopodial length of ~250% (Fig. 4f). These results suggest that cells within a polarized epithelial sheet, which maintain correct cell shape, cell–cell junctions, and cell-basal lamina contacts, are more recalcitrant to photomanipulation than delaminated cells.

When photoactivating dominant negative PA-RacT17N expressing cells the abnormal protrusion morphology observed prior to photoactivation was unaffected (Fig. 5a) however protrusion dynamics were significantly reduced (Fig. 5b,c). This effect on protrusion dynamics was only observed during photoactivation (Fig. 5b) and this effect was specific to PA-RacT17N expressing cells (protrusion dynamics were unaffected when photoactivating the constitutively active or photo-insensitive constructs, see Fig. 5c). These morphologically abnormal protrusions with very limited dynamics resemble in many ways the abnormal basal protrusions observed in *lgl* mutant cells (Fig. 2e, Supplementary Fig. 2 and Supplementary Movie 2).

To ensure that the observed effects were not simply due to the photoactivation protocol itself, we photoactivated cells that were expressing the photo-insensitive version of the transgene, or simply an mCherry reporter. In both cases lamellipodial area and filopodial length were unaffected by the expression of these constructs or by the process of photoactivation (Supplementary Fig. 4e) suggesting that the observed effects on protrusion morphology and dynamics are due to an alteration in Rac activity.

## Discussion

The seminal experiments of Hall and colleagues in the early 90s (refs. 34–36) suggested that Rac and Cdc42 act in parallel pathways to induce lamellipodia and filopodia respectively. In the intervening years mounting evidence has suggested a far more complex scenario, a scenario where different Rho GTPases regulate one another in a tight spatiotemporal manner at the leading edge[37] and where downstream effectors from several Rho GTPases are required for lamellipodial formation[38,39]. There is further evidence to suggest that lamellipodia could represent an intermediate in filopodial formation, with filopodia emerging from the lamellipodial F-actin network[40]. We provide further evidence to support this idea in an *in vivo* system, as we observe filopodia emerging from the lamellipodial sheet as the lamellipodium contracts when increasing Rac activity following photoactivation (Fig. 4b). However, despite this evident complexity, it is still widely accepted that the activity of different Rho GTPases is required to form different actin structures. It is likely that in order for a cell to form a certain type of protrusion at the correct location, the temporal and spatial regulation of Rho GTPase activity, and consequently of their effectors, will be highly complex. However we provide here evidence to suggest that the key determinant of protrusion form is not complex, it is simply the local level of Rac activity. We have previously shown that we can rescue dynamic protrusions, including filopodia, in *cdc42* mutant cells by overexpressing activated Rac in these cells[5]. Here we show that by locally increasing Rac activity we can convert lamellipodial protrusions into filopodial protrusions, both in delaminated cells and in polarized cells within an epithelial sheet. Thus we believe that the level of Rac activity is a key factor in determining protrusion form.

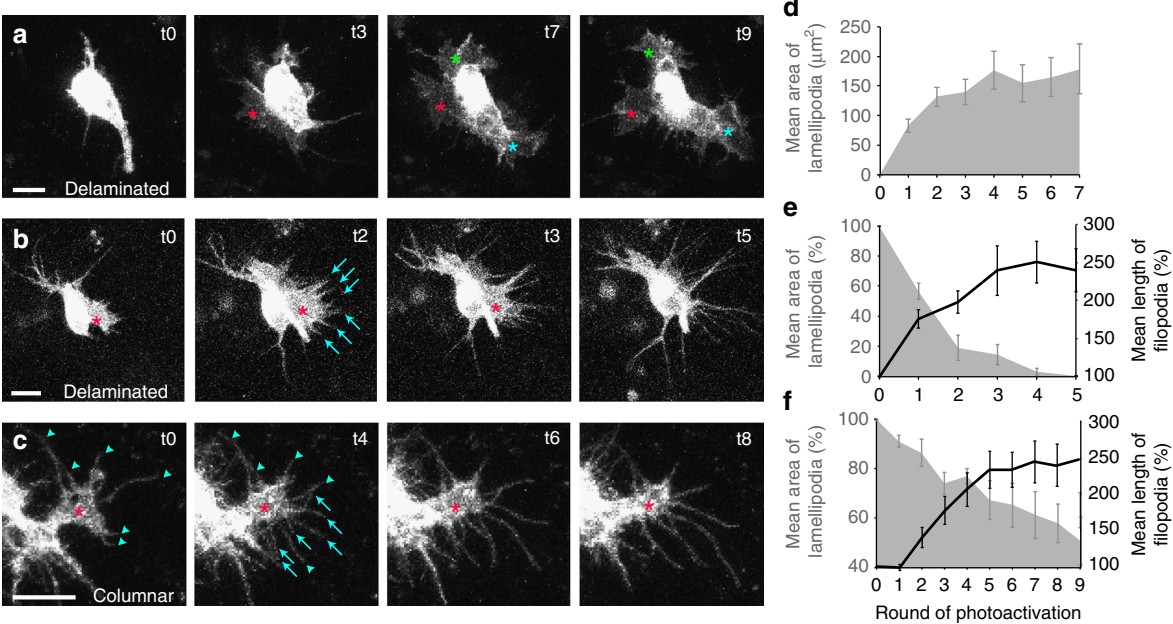

**Figure 4 | The level of Rac activity determines protrusion form.** (**a–c**) Live imaging of MARCM clones of epithelial cells overexpressing a constitutively active form of photoactivatable Rac, PA-RacQ61L (stills taken from Supplementary Movies S4–S6). (**a**) A delaminated epithelial cell rapidly develops three prominent lamellipodia upon photoactivation (asterisks). (**b**) A delaminated epithelial cell has a lamellipodium prior to photoactivation (asterisk). Upon photoactivation, the lamellipodium progressively retracts and new filopodia develop and grow (arrows). (**c**) A polarized epithelial cell within the epithelial sheet that possesses a basal lamellipodium (asterisk) as well as six pre-existing filopodial protrusions (arrowheads). Upon photoactivation, the lamellipodium slowly retracts and new filopodia develop and grow (arrows). (**d–f**) Quantification of mean lamellipodial area (grey area) and mean filopodial length (black line) over time, prior to photoactivation (t0) and after each consecutive round of photoactivation (t1–t9). (**d**) Delaminated cells with no lamellipodia prior to photoactivation ($n = 10$ new lamellipodia following photoactivation). (**e**) Delaminated cells with prominent lamellipodia prior to photoactivation ($n = 8$ lamellipodia, $n = 26$ new filopodia following photoactivation). (**f**) Polarized epithelial cells within the epithelial sheet with prominent lamellipodia prior to photoactivation ($n = 12$ lamellipodia, $n = 82$ new filopodia following photoactivation). Error bars represent s.e.m. Scale bars, 10 µm.

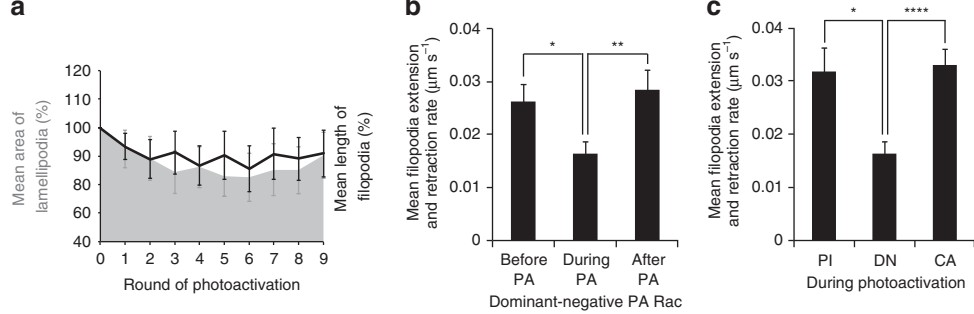

**Figure 5 | Photoactivation of dominant negative PA-Rac reduces protrusion dynamics.** (**a**) Quantification of mean lamellipodial area (grey area) and mean filopodial length (black line) over time, prior to photoactivation (t0) and after each consecutive round of photoactivation (t1–t9) in cells expressing the dominant negative form of PA-Rac. The size of both does not significantly change upon photoactivation (t0, before photoactivation, is compared to each of the nine rounds of photoactivation t1–t9, using Student's t-test, $n = 10$ lamellipodia, $n = 32$ filopodia). (**b**) Mean extension and retraction rates of filopodia from cells expressing a dominant negative form of PA-Rac ($n = 31$ filopodia) before, during and after photoactivation. (**c**) Mean extension and retraction rates of filopodia from cells expressing a photo-insensitive (PI, $n = 66$ filopodia), a dominant negative (DN, $n = 34$ filopodia) and a constitutively active (CA, $n = 88$ filopodia) form of PA-Rac during photoactivation. Error bars represent s.e.m. Student's t-test was performed to determine statistical significance and P values are shown on graph. $P > 0.5$ was considered not significant, $*P < 0.05$, $**P < 0.01$, $***P < 0.001$, $****P < 0.0001$.

Our identification of a gradient of Rac activity along the apicobasal axis of polarized epithelial cells explains the graded distribution of Rac-induced protrusions within these cells. We have shown that this gradient can be modified in cells mutant for apicobasal polarity regulators and that this modification consequently affects the form and position of protrusions within the basolateral domain of these cells. This therefore provides a mechanism whereby polarity proteins can spatially regulate Rac activity and consequently influence actin cytoskeletal dynamics within an epithelial sheet. This can ensure that every cell within a polarized sheet not only maintains a correct cell shape, but also forms and positions its protrusions correctly. We have previously observed that when we genetically increase Rac activity (by overexpressing sif/TIAM1, or constitutively active Rac, or in *baz* mutant cells) filopodia are no longer restricted to the base of the cell, but are found to occur all along the lateral membrane, occurring at the expense of the lateral sheet-like protrusions[5]. These results perfectly fit our model of a Rac activity gradient that

is required to position lateral protrusions and restrict filopodia to the base of the cell.

The existence of actin-based cell protrusions have previously been reported in *Drosophila*, both in the notum[5,41,42] and in other epithelia[43–45]. More recently, cell protrusions have been observed in epithelial cells in other organisms, including the epithelial cells of the somites of chicken embryos[46], in the embryonic epidermis of *Caenorhabditis elegans*[47] and during epithelial fusion in mouse embryos[48]. In each case, these actin-rich protrusions were found to be dependent on Rac activity. It is therefore tempting to postulate that a gradient of Rac activity could be a mechanism employed by several types of epithelia. Cells within three-dimensional epithelial sheets that possess a polarized distribution of protrusions could use a gradient of Rac activity to correctly determine protrusion position along the apicobasal axis of the cell, just as we have observed in the notum of the fly.

As we have shown previously, the basal protrusions in the epithelial cells of the notum are involved in cell–cell communication events, specifically basal filopodia are required to propagate Delta-Notch signalling in this epithelium[9]. Therefore it is likely that any disruption to cell polarity will affect the Rac activity gradient and have profound consequences, not only on cell shape and protrusion positioning, but also on cell signalling within the epithelial sheet. It has long been known that a loss of apicobasal polarity is a prerequisite for epithelial-to-mesenchymal transitions, and we additionally show here that an increase in Rac activity leads to an increase in these invasive events. Rac1 is over-expressed in many tumour-types and there is accumulating evidence to suggest that Rac1-dependent cell signalling is important for tumour progression[7,49]. This work therefore provides a mechanism through which polarity proteins can influence Rac activity, thereby maintaining correct cell shape and epithelial integrity.

## Methods

**Transgenic *Drosophila* stocks and crosses.** Fly stocks were raised on standard medium at 25 °C. The following stocks were used: UAS-Rac-FRET (on II #32050 or III #31431), *sif*[ES11]/TM6B (#9126), Df[3L]CH18/TM3 (#6463), UAS-mCherry-PA-RacQ61L (#32049), UAS-mCherry-PA-RacT17N (#31429), UAS-mCherry-C450M-PA-RacQ61L (#32048), FRT82b TubP-Gal80 (#5135), Ubx-FLP (#42718), Pnr-Gal4 (#25758), FRT82b (#2051), FRT19A (#1709), *lgl[4]* FRT40A/CyO (#36289), *baz[4]* FRT9-2/FM7c (#23229), TubP-Gal80 FRT40A (#5192), hsflp TubP-Gal80 FRT19A (#5132) and *dlg1[14]* FRT101/FM7a (#36283) were obtained from Bloomington *Drosophila* Stock Center; UAS-mCherry::CAAX (#109594) and ubi::E-cad-GFP (#109007) were obtained from *Drosophila* Genetic Resource Center (DGRC). Resille-GFP (also known as P{PTT-un1}CG8668[117-2]) was obtained from E. Wieschaus. Ubi-nls-GFP, FRT 19A and Ubi-nls-GFP, FRT 40A are lab stocks.

The following recombinations were performed to generate new stocks: Pnr-Gal4 and FRT82b on III; Neu-Gal4 and *sif*[ES11] on III; *dlg1[14]* and FRT19A on I, and then *dlg1[14]*, FRT19A and Ubx-FLP on I; *baz[4]* FRT9-2 and Ubx-FLP on I.

**Dissections and live imaging.** Nota from pupae 12–16 h APF were dissected in PBS and the tissue fixed in 4% formaldehyde for 20 min, before being permeabilised with PBS containing 0.1% Triton X-100.

For live imaging, animals with the appropriate genotype were prepared by cutting a window in the pupal case, attached to a slide with double-sided sticky tape. A coverslip with a drop of injection oil was then placed over the notum, supported by coverslips at either end to allow imaging on inverted confocal microscopes.

**Immunocytochemistry.** We used the following primary antibodies at the indicated dilutions for this study: rat anti-E-Cad [1:100, DSHB (DCAD2)], rabbit anti-Baz (1:2,000, gift from A. Wodarz[50]). Secondary antibodies from Molecular Probes were Alexa Fluor 488, 546 and 633. Images were acquired on a Zeiss LSM880 confocal, and assembled using Adobe Photoshop.

**FRET experiments.** To analyse Rac activity levels along the apicobasal axis of living epithelial cells, Neu-Gal4 was used to drive expression of the UAS-Rac-FRET biosensor in the sensory organ precursor cells of the fly notum in living pupae. For analysis of Rac activity in *sif* mutant cells the following genotype was used: UAS-Rac-FRET:Neu-Gal4 *sif*[ES11]/Df[3L]CH18. For analysis of Rac activity in

*baz*, *dlg* or *lgl* mutant cells, MARCM was used to generate clones of mutant tissue within which only the sensory organ precursor cells expressed UAS-Rac-FRET. The following genotypes were imaged: for *baz* mutants, Ubx-FLP, *baz[4]*, FRT9-2/TubP-Gal80, FRT9-2; UAS-Rac-FRET/Neu-GAL4; for *dlg* mutants, Ubx-FLP, *dlg1[14]*, FRT19A/TubP-Gal80, FRT19A; UAS-Rac-FRET;Neu-Gal4, or Ubx-FLP, *dlg1[14]*, FRT19A/TubP-Gal80, FRT19A; UAS-Rac-FRET/Neu-Gal4; and for *lgl* mutants, Ubx-FLP; *lgl[4]*, FRT40A/TubP-Gal80, FRT40A; Neu-Gal4/UAS-Rac-FRET.

Crosses were performed at 25 °C, pupae were placed at 18 °C at 0 h APF overnight, then shifted to 25 °C for at least 2 h prior to imaging at 12–15 h APF, to allow sufficient expression of the Rac-FRET biosensor. Rac-FRET images were acquired with a Leica SP2 inverted confocal microscope equipped with a × 40/1.25 NA oil objective with PL APO correction and using a × 2.5 optical zoom. Samples were excited with a 458 nm laser, and CFP and YFP emission signals were collected simultaneously through channel I (470–510 nm) and channel II (525–600 nm), respectively. For all normal acquisitions, z-series were acquired from the bottom to the top of the cells using 1 μm z-sectioning. For the photobleaching control 1, this normal acquisition method was immediately repeated for the same field of view. For the photobleaching control 2, the z-series was instead acquired from the top to the bottom of the cells within a different field of view, but from the same animals.

FRET efficiencies were calculated using Fiji software. Using the freehand selection tool, individual cells were drawn around and then within this whole region of interest the mean grey value for both the YFP and CFP channels was obtained. For each cell, this process was repeated at each z-slice from apex to base. The corresponding background values were then also obtained for each z-slice. These values were exported to excel software where subsequently background-subtracted YFP/CFP ratios were calculated for each z-slice, followed by subtraction of the recommended bleed-through constant (0.6, see ref. 51) to account for non-FRET associated YFP signal. To facilitate the comparison of cells with differing cell lengths, z-series were then divided into five sections and the mean FRET value across the z-planes in each section was calculated, for example: a cell with ten z-slices would have two z-slices averaged per section. These five sections correspond to sections 1–5 on the FRET figure x-axes, with section 1 being the most apical and section 5 the most basal. Following this sectioning process, the FRET values for each section were then normalized by subtracting the section 1 (apical) value, making section 1 a value of 0 for all cells. Mean FRET values at each 'section' were then calculated for each genotype. This methodology allowed us to assess intracellular FRET variation along the apical–basal cell axis, and to reliably compare this data between different cells, as well as between genotypes.

Ratiometric FRET images were generated using Fiji software. Firstly, the same cropped area for the cell of interest was applied throughout the z-series. Next, the background was calculated and subtracted at each z-slice for both the YFP and CFP channels. Then the YFP images were registered to the CFP images to ensure correct alignment of pixels. CFP images were then converted to 32-bit. A smooth filter was applied to both the CFP and YFP images, and finally the YFP images were equally thresholded, prior to generation of the ratio images using the Calculator Plus processing tool. For optimal visualization of the ratiometric images, the maximum displayed value was adjusted accordingly (Image > adjust > Brightness/Contrast > Set). Mean ratio values were obtained from the ratiometric images by measuring the mean grey value within particular regions of interest.

**Photoactivation.** Pupae were aged at 18 °C in the dark overnight and then shifted to 25 °C to increase transgene expression for a couple of hours prior to mounting. Mounting was performed using a red light source to minimize photoactivation prior to imaging. Photoactivation and time-lapse imaging were performed using a Zeiss LSM5 Exciter AxioObserver inverted confocal microscope equipped with an EC Plan-NeoFluar × 40/1.30 oil lens and Zen acquisition software. mCherry was excited with a 543 laser and emission was split through HFT458/543/633 and LP560 filters. For photoactivation, a 458 nm laser was used at 100% capacity, the bleaching speed allowed for a 1.60 μm pixel dwell and we did 50 iterations per round of photoactivation.

Typically, we imaged the cells for 5 min (five z-scans of 10–15 μm, every 1 μm, per minute) before photoactivating. Then we imaged the cells for additional 10 min (ten similar z-scans) while photoactivating after each of the ten z-scans. Finally, cells were again imaged for 5 min (five z-scans) with no photoactivation.

Time-lapse movies were analysed using Fiji software. The area of individual lamellipodia and length of individual filopodia were traced manually in each time point using the 'polygon selection' and the 'segmented line' tools respectively. The number of lamellipodia per cell were counted manually. Genotypes imaged in the photoactivation experiments were:

Ubx-FLP/+; UAS-mCherry-PA-RacQ61L/+; FRT Pnr-GAL4/FRT TubP-Gal80.

Ubx-FLP/+; UAS-mCherry-PA-RacT17N/+; FRT Pnr-GAL4/FRT TubP-Gal80.

Ubx-FLP/+; UAS-mCherry-C450M-PA-RacQ61L/+; FRT Pnr-GAL4/FRT TubP-Gal80.

Ubx-FLP/+; UAS-mCherry::CAAX/+; FRT Pnr-GAL4/FRT TubP-Gal80.

**Calculations and statistical analysis.** Excel was used to perform calculations, generate most graphs and calculate statistical significance with Student's t-test,

where $P > 0.5$ was considered not significant, $*P < 0.05$, $**P < 0.01$, $***P < 0.001$, $****P < 0.0001$. Calculation of apical and junctional length was performed using Volocity software. Dot plots were generated using R software.

**Data availability.** The data that support the findings of this study are available from the corresponding author upon request.

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

## Acknowledgements

We wish to thank the fly community for their generosity with reagents, especially the Tsien and Wieschaus Labs for fly stocks and the Wodarz lab for Baz antibody. We thank members of the lab and the School of Life Sciences for advice and support. We especially acknowledge School of Life Sciences imaging (SLIM) for invaluable help with the confocal microscopes and Angeliki Malliri, Andrew Porter, Andrew Renault and Peter Shaw for critical reading of the manuscript. This work was supported by Cancer Research UK [grant numbers C36430, A12891]. A.C. and N.M. were supported by a Cancer Research UK Career Establishment Award to M.G.; L.F. was supported by a Nottingham Vice-Chancellor's Scholarship for Research Excellence Award.

## Author contributions

M.G., A.C., N.M. were responsible for experimental design; M.G., A.C., N.M., L.F. carried out experiments and analysed data; M.G. devised and supervised the project, and wrote the manuscript with input and revisions from A.C. and N.M.

## Additional information

**Competing interests:** The authors declare no competing financial interests.

