## [Peer Review File · Nature Communications]

Reviewers' comments:

Reviewer #1, expert in epithelial polarity and GTPases (Remarks to the Author):

A localised Rac gradient is thought to be established by regulators associated with cell-cell junctions that either directly regulate Rac activity or that modulate the activity of Rac GEFs and GAPs. The gradient is believed to be important for junction maturation and polarization of epithelial cells. This paper builds on this work by investigating whether the same Rac gradient also controls the formation of distinct actin-driven cell protrusions along the lateral membrane in the *Drosophila* pupal notum.

The paper asks an interesting question and shows a considerable amount of data supporting the conclusion. However, there are considerable gaps and some of the data are confusing.

1) A single cell type is studied but a general model is proposed. Are similar Rac-dependent processes observed in other types of epithelial cells?

2) Figure 1:

Why is the gradient the same in *sif/tiam1* mutants and in *baz* mutants? One would expect increased levels in the latter if the model is correct as *baz* is supposed to function as an inhibitor of *tiam*.

The FRET images are confusing. *Tiam1* is a lateral protein that that activates Rac at the plasma membrane. Why is there no enrichment of active Rac at the membrane? Why is the signal mostly cytosolic? What does the quantification show, total activity in a section or lateral/membrane-associated Rac?

3) Figure 1A: While there might be an actin-rich apical membrane in these cells, it is impossible to make out microvilli or, even more challenging, decide whether they changed in response to manipulation of *baz* or *sif*. This is a typical case requiring electron microscopy. It should also be clarified if these cells indeed polarize in a *baz*-independent manner.

4) Figure 2: Microvilli and basal protrusions are concluded to be morphologically normal in *dlg* mutants; however, there are no data shown that support such a claim.

- As in figure 1, specific statements about the structure of the apical membrane are not possible based on these confocal sections. Moreover, A and B do not look very similar and panel C seems to have increased apical staining (and is overexposed).

- Basal filopodia seem to be longer in *lgl* cells and in *dlg* they seem shorter. The length of basal protrusions should be quantified. These images don't suggest that *dlg* leads to increased number of basal protrusions as suggested in the quantification but to shorter ones. Hence, one wonders how specific the effect is for lateral protrusion.

- panels F and G lack statistical tests. The two distributions in G don't appear to be very different.

5) Figure 3: it should be made clear that there is no effect on apicobasal polarity in cells mutant in *lgl* and *dlg*. There is a minor expansion of the junctional area but no expansion of the apical domain as predicted. Why do the authors think these cells respond so differently to the inactivation of polarity determinants to other epithelial cells. How comparable are these cells with other types of epithelia? (see also comments 1 and 3)

6) Extended Figure 3 lacks an image from control cells. How can one see that *Baz* is distributed any differently than in controls?

7) Figure 4 is unconvincing. It shows that a strong increase in Rac activity induces actin protrusions, which is already known. Two of the cells show are delaminated and it is not clear how they should support the conclusion about lateral protrusions. Of the third columnar cell one sees a corner that seems to be the base of the cell which has long protrusions anyway. How does this

figure support the existence of a Rac gradient that controls protrusion length along the lateral membrane? A system allowing more specific mid-lateral Rac activation would be required but that seems technically rather challenging. Perhaps stimulating Rac by overexpressing sif/tiam might be a more promising approach.

Reviewer #2, expert in FRET biosensors (Remarks to the Author):

In the present paper, Couto et al. have attempted to demonstrate that the gradient of Rac activity within epithelial cells determine the form and space of membrane protrusions. The authors use *Drosophila pupal notum* as the model system, a Rac FRET biosensor, and photo-activatable Rac proteins for this purpose. First, they demonstrated that the Rac activity forms a gradient; high at the base and low at the apex of the cell. They next show that in the *sif*, *dlg*, or *lgl* mutant, both the gradient and membrane protrusions are marked perturbed. Finally, it has been shown that photo-induced activation of Rac increases in lamellipodial protrusions. Most data appear to be solid. The references are cited in proper quantities. Use of standard error obscures the cell-to-cell variety, but generally statistics are used properly. Overall this work has been well done in an organized way, written clearly, and adds new insight into the role of Rac gradient in the form and space of lamellipodia in the epithelial cells. Most conclusions are supported by the data. If the gradient of Rac activity could be shown in the X-Z plane, the proposal could be more easily understood.

Major comments:

1. Localization of the biosensor is not clearly described. Rac1 is supposed to locate primarily at the plasma membrane. Is the biosensor also localized to the plasma membrane? If not, what does the biosensor measure? In any event, either CFP or YFP image of the biosensor should be shown.
2. Are there any variations among the pupae? In all experiments, the data are summed up and shown by mean and standard error. It would be informative if the difference among the sample is also shown.
3. Re: Difference in Rac1 activity among the wild-type and mutants: This could be most easily and impressively represented if the XY images of Rac1 activity of chimera can be shown as in Fig. 3C and 3D. Although the data are normalized to the values of apical region of the cell, there should be XY planes that show the difference.

Minor comments:

1. Fig. 2F, G: This may be specific to the reviewer's PC, but the panels are overlain by black sheets.

Response to reviewers.

First we would like to thank the reviewers for reviewing our paper. Having revised the manuscript to take the reviewers comments into account, we believe we have a stronger paper and we thank the reviewers for their input. Our point-by-point response is detailed below.

Reviewer #1:

1) A single cell type is studied but a general model is proposed. Are similar Rac-dependent processes observed in other types of epithelial cells?

This is an excellent question, and one that we should have addressed in our discussion. Rac dependent processes have been observed in several types of epithelia, most extensively in *Drosophila* (in the notum, wing imaginal disc and the abdominal epidermis e.g. Renaud and Simpson, Dev. Biol., 2001; De Joussineau et al., Nature, 2003; Hsiung et al., Nature, 2005; Demontis and Dahmann, Dev. Dyn., 2007; Bischoff et al., Nature Cell Biology, 2013) and more recently in other organisms: for example, in epithelial cells of the somites of chicken embryos (Sagar et al., Development, 2015), in the embryonic epidermis of *C. elegans* (Walck-Shannon et al., Development, 2015), during epithelial fusion in mouse embryos (Rolo et al., Elife, 2016). Given the wide variety of epithelia within which these protrusions have been identified, we believe the tone of our article is appropriate. We have now discussed this point in our Discussion (see page 14).

2) Figure 1:

Why is the gradient the same in *sif/tiam1* mutants and in *baz* mutants? One would expect increased levels in the latter if the model is correct as *baz* is supposed to function as an inhibitor of *tiam*.

We get a similar gradient in *sif/tiam1* mutants and in *baz* mutants as this data has, in both cases, been normalised to the apical FRET value for each cell. Therefore, we are measuring the difference in Rac activity along the apicobasal axis, not absolute values of Rac activity. As a result, high levels of Rac activity along the apicobasal axis (as we might predict in a *baz* mutant cell) or low levels of Rac activity throughout the cell (as we might predict in a *sif* mutant cell) would both result in the same disruption to the Rac gradient, i.e. a shallowing, or loss of, the Rac activity gradient. The data shown supports this, as in both mutants a much shallower gradient is observed compared to the wild type situation. We thank the reviewer for pointing this out and now have made it more explicit that these data are normalized to the apical value by stating this in the figure legends, as well as in the Methods.

The FRET images are confusing. Tiam1 is a lateral protein that that activates Rac at the plasma membrane. Why is

there no enrichment of active Rac at the membrane? Why is the signal mostly cytosolic?

We appreciate that the membrane enrichment of Rac activity is difficult to see in Figs 1F-F'''' due to the scale of the images. We do however show this membrane enrichment more clearly in Supplementary Figure 1C. This is also referred to in the main text (page 6): "However, as expected, within a single confocal section, we observed a higher FRET signal and therefore higher levels of Rac activity within protrusions and at the cell cortex (Supplementary Figure 1C-D)." More areas of white and red (high Rac activity) can be seen at the membrane and in particular within the membrane protrusions compared with cytoplasmic regions. The numbers shown underneath denote the average FRET ratio within those specific regions, confirming this enrichment. It should also be noted that the FRET probe being used is not tethered to the membrane and therefore localisation within the cytosol is expected. However, we expect that it will mostly be activated at the membrane, and this is supported by the images shown in Supplementary Figure 1C. In response to the reviewers comment, we have added an additional panel to Supplementary Figure 1 (Supplementary Figure 1D) that more clearly shows high levels of Rac activity at the cell membrane and in protrusions.

What does the quantification show, total activity in a section or lateral/membrane-associated Rac?

For the quantification, total activity in a section was measured. Due to the FRET probe not being membrane-tethered, it would be difficult to accurately ascertain membrane-only measurements at all apical-basal locations. We have revised the corresponding methods section to make this clearer (page 17).

3) Figure 1A: While there might be an actin-rich apical membrane in these cells, it is impossible to make out microvilli or, even more challenging, decide whether they changed in response to manipulation of baz or sif. This is a typical case requiring electron microscopy. We appreciate that the images are of insufficient resolution to rule out any changes to the apical protrusions in the mutants analysed. The reviewer makes a good suggestion in the use of electron microscopy to study these protrusions in greater detail. We do however feel that the study of these apical protrusions is beyond the scope of this paper, especially since the formation of these apical microvillar-like protrusions is not dependent on Rac activity. These apical protrusions form in all mutants analysed, including cells mutant for the three *Drosophila* Racs, when overexpressing dominant negative Rac, and in cells mutant for *cdc42* or *scar* (see Georgiou and Baum, 2010). Since this paper focuses on Rac activity levels and their importance in regulating protrusion form and position along the apicobasal axis, we believe that the study of these apical protrusions, which do not rely on Rac activity, would be better served in a separate paper.

We have however taken the reviewers comments on board and now no longer refer to these protrusions as 'microvilli', rather we call them 'apical microvillar-like protrusions'. We feel this is justified, as electron microscopy studies of other *Drosophila* chitin-synthesizing epithelia, like the midgut and the embryonic epidermis, have described the presence of either microvilli or microvillar-like structures (reviewed by Moussian, Insect Science, 2013).

It should also be clarified if these cells indeed polarize in a *baz*-independent manner.

This has been shown in previous publications. In both Georgiou et al., 2008 and Georgiou and Baum, 2010, we show that *baz* mutant cells polarise normally and maintain a columnar cell shape. We do however show that in *baz* mutant cells, filopodia are no longer restricted to the base of the cell. Rather filopodial protrusions are observed throughout the basolateral domain, and occur at the expense of intermediate level sheet-like protrusions. The same phenotype is observed when overexpressing activated Rac in these cells (Georgiou and Baum, 2010) supporting the notion that high levels of Rac activity promotes filopodia formation. This is discussed on page 5.

4) Figure 2: Microvilli and basal protrusions are concluded to be morphologically normal in *dlg* mutants; however, there are no data shown that support such a claim.

- As in figure 1, specific statements about the structure of the apical membrane are not possible based on these confocal sections. Moreover, A and B do not look very similar and panel C seems to have increased apical staining (and is overexposed).

We take the reviewers comments on board and no longer state that the apical microvillar-like structures are 'morphologically normal', we now state that these actin rich apical protrusions are 'still present and dynamic'. See page 7.

We agree with the reviewer that panel C in Figure 2 was overexposed. As a result we have included an alternative *lgl* mutant cell, and replaced panels C-C''.

- Basal filopodia seem to be longer in *lgl* cells and in *dlg* they seem shorter. The length of basal protrusions should be quantified. These images don't suggest that *dlg* leads to increased number of basal protrusions as suggested in the quantification but to shorter ones. Hence, one wonders how specific the effect is for lateral protrusion.

We thank the reviewer for this suggestion. We have now included quantification of basal protrusion length, and included this in Supplementary Figure 2. This shows that *dlg* basal protrusions are similar in length to wild type basal protrusions, but *lgl* basal protrusions are significantly longer. In summary, *dlg* mutant cells have lost intermediate protrusions but have basal protrusions

that are wild type in terms of their morphology and dynamics.

- panels F and G lack statistical tests. The two distributions in G don't appear to be very different. Statistical tests have now been carried out and are shown in Figure 2F-G. These show a significant reduction in the number of intermediate level protrusions in both *dlg* and *lgl* mutant cells, when compared to wild type.

5) Figure 3: it should be made clear that there is no effect on apicobasal polarity in cells mutant in *lgl* and *dlg*. There is a minor expansion of the junctional area but no expansion of the apical domain as predicted. We do not consider that there is no effect on apicobasal polarity in *lgl* and *dlg* mutant cells, however we thank the reviewer for this comment as we now realise that we failed to point out in the original manuscript that there is a significant expansion of the apical domain (in addition to the junctional domain) in *lgl* mutant cells. This has now been rectified (see page 8 and Figure 3B). Secondly, we do not consider that the expansion of the junctional domain is minor as, in the case of *dlg*, there is an expansion of 175%. Additionally, we were not surprised to see an effect on the junctional domain in these mutants, as both apical and junctional proteins have previously been shown to be basally mislocalised in these mutants (Bilder et al., Science, 2000) and an expansion of the junctional domain has previously been shown in *lgl* mutant clones in the pupal eye retina (Grzeschik et al., Developmental Biology, 2007). Again, we thank the reviewer for making this point as we have now modified the text to make this clearer and to acknowledge this previous work (see page 7).

Why do the authors think these cells respond so differently to the inactivation of polarity determinants to other epithelial cells. How comparable are these cells with other types of epithelia? (see also comments 1 and 3)

This is a good question. It is known that polarity protein mutants, and especially those of the scribble complex, behave very differently if the whole tissue is mutant, or if small mosaic clones are generated, surrounded by wild type tissue. The phenotype in the former is dramatic, with a loss of polarity, multilayering and tissue overgrowth (Bilder and Perrimon, Nature, 2000; Bilder et al., Science, 2000). However, phenotypes observed when generating small mutant clones are generally mild. For example, in Grzeschik et al. 2007 (Developmental Biology 311, pp106-123) the authors state that *lgl* clones in the larval eye disc do not show cell polarity defects and show relatively minor polarity defects at pupal stages. We therefore believe that the pupal notum is comparable to other types of epithelia, at least in *Drosophila*.

6) Extended Figure 3 lacks an image from control cells. How can one see that Baz is distributed any differently than in controls?

We thank the reviewer for this comment. A new Supplementary Figure 3 has now been generated, showing neighbouring wild type tissue. The new figure shows expanded junctions in *dlg* mutant cells, with Baz colocalising with E-Cad throughout the entire junctional domain.

7) Figure 4 is unconvincing. It shows that a strong increase in Rac activity induces actin protrusions, which is already known. Two of the cells shown are delaminated and it is not clear how they should support the conclusion about lateral protrusions. Of the third columnar cell one sees a corner that seems to be the base of the cell which has long protrusions anyway. How does this figure support the existence of a Rac gradient that controls protrusion length along the lateral membrane?

The focus of Figure 4 is not to support the existence of a Rac gradient per se, it is rather to support the underlying hypothesis that it is the level of Rac activity that determines protrusion form. The reviewer is right to point out that an increase in Rac activity in cultured cells has been shown to induce de novo formation of actin-rich protrusions. In figure 4A and 4D we show that we can achieve the same result in our model system, which is a living organism. We believe this is a considerable achievement in itself and a step forward in the understanding of protrusion formation, as the intact *Drosophila* epithelium provides a context that is lacking in any cultured system.

However, the de novo formation of protrusions upon Rac activation is not the main conclusion drawn from our experiments. Figures 4B, 4C, 4E and 4F illustrate how a further increase in Rac activity transforms lamellipodia into filopodia. To our knowledge, this phenomenon has not been shown in any published work, and therefore our conclusions are completely novel.

The reviewer is unconvinced of the suitability of the delaminated cells in our study. Delaminated cells have lost their apicobasal polarity and therefore form both lamellipodia and filopodia irrespective of the cell's x-z axis. In our study we show that lamellipodia within these delaminated cells transform into filopodia upon Rac activation (Figure 4B, 4E). This is in addition to similar observations that we report within columnar cells with intact apicobasal polarity (Fig 4C, 4F). The data we have obtained using delaminated cells supports our conclusion that it is the level of Rac activity that determines protrusion form, insofar as intermediate levels of Rac activity promote the formation of lamellipodial sheets and high levels of Rac activity promote the formation of filopodia.

The reviewer is concerned that the columnar cell shown in Figure 4C has long filopodia at the base prior to photoactivation. It is expected to see filopodia at the base of the cell, more so when overexpressing

constitutively active PA-RacQ61L. Figure 4C depicts a large lamellipodium at the base of a columnar cell (marked with an asterisk). Before photoactivation (t_0), this lamellipodium has 2 long and 4 short finger-like protrusions. Upon photoactivation, we can see how 2 of the small finger-like protrusions progressively extend and how several new protrusions appear and progressively extend from the lamellipodium. Most importantly, this filopodial extension is accompanied by the simultaneous retraction of the lamellipodial sheet. This progressive increase in the total length of filopodia and progressive decrease of lamellipodial area is consistently observed, and quantified in Figure 4F.

We would like to thank the reviewer for this comment, as we have changed the labelling of these protrusions in a modified Figure 4. We hope that this new labelling will more clearly highlight pre-existing filopodia (which are now labelled with arrowheads) and newly formed filopodia (now marked with arrows).

Please note that in the original submission, Panel F of Figure 4 may not have displayed correctly (the grey area showing lamellipodial retraction does not come up on all of our computers). This was probably due to converting these Illustrator files to PDFs and this might have added to the confusion when interpreting these results. We are now supplying the original Illustrator files, so hopefully this will not happen again.

A system allowing more specific mid-lateral Rac activation would be required but that seems technically rather challenging.

The reviewer is correct in stating that activating Rac and observing its effect in the intermediate level protrusions would be technically challenging. Sheet-like protrusions at an intermediate level expand through several microns on the z axis and therefore it is not possible to photoactivate and image the consequences using a regular confocal microscope, which focuses on a particular z plane. However, we do agree that photoactivating Rac at an intermediate level would be informative and we are currently preparing a grant application to acquire funds to use a light-sheet microscope, which would allow us to photoactivate and observe its effect on protrusion morphology along the x-z axis.

Perhaps stimulating Rac by overexpressing *sif/tiam* might be a more promising approach.

This experiment has been done and supports our conclusions. We have previously shown, in Georgiou and Baum, 2010, that the overexpression of *sif/tiam* phenocopies *baz* mutant cells as well as cells overexpressing constitutively active Rac. The result for each of these genotypes is that filopodia are no longer restricted to the base of the cell, but are found to occur all along the lateral membrane, occurring at the

expense of the lateral sheet-like protrusions. These experiments are now discussed on page 14.

Response to Reviewer #2:

In the present paper, Couto et al. have attempted to demonstrate that the gradient of Rac activity within epithelial cells determine the form and space of membrane protrusions. The authors use *Drosophila pupal notum* as the model system, a Rac FRET biosensor, and photo-activatable Rac proteins for this purpose. First, they demonstrated that the Rac activity forms a gradient; high at the base and low at the apex of the cell. They next show that in the *sif*, *dlg*, or *lgl* mutant, both the gradient and membrane protrusions are marked perturbed. Finally, it has been shown that photo-induced activation of Rac increases in lamellipodial protrusions. Most data appear to be solid. The references are cited in proper quantities. Use of standard error obscures the cell-to-cell variety, but generally statistics are used properly. Overall this work has been well done in an organized way, written clearly, and adds new insight into the role of Rac gradient in the form and space of lamellipodia in the epithelial cells. Most conclusions are supported by the data. If the gradient of Rac activity could be shown in the X-Z plane, the proposal could be more easily understood.

We agree that showing Rac activity in the x-z plane would be ideal, but this would be extremely technically challenging. As mentioned above, in reply to a point from Reviewer #1, we are currently preparing a grant application to apply for funding that would allow us to use a light-sheet microscope, which would allow us to image these cells in the x-z plane.

Major comments:

1. Localization of the biosensor is not clearly described. Rac1 is supposed to locate primarily at the plasma membrane. Is the biosensor also localized to the plasma membrane? If not, what does the biosensor measure? In any event, either CFP or YFP image of the biosensor should be shown.

We thank the reviewer and have now more clearly described the source and localisation of the biosensor (see page 6 in modified manuscript). We purposely chose to use a well-characterised biosensor that has been shown to work previously in multiple in vivo systems (see references 21 and 22 in the manuscript). The FRET probe being used is a modified version of Raichu-Rac and is not tethered to the membrane and therefore localisation within the cytosol is to be expected. However, we expected that it will mostly be activated at the membrane, and that is what we found (see Supplementary Figure 1C). We have also added an additional panel to Supplementary Figure 1 (Supplementary Figure 1D) that more clearly shows high levels of Rac

activity restricted to the cell membrane and in protrusions.

A CFP and YFP image of the biosensor was included (Supplementary Figure 1B) which shows a uniform subcellular localisation of the protein, and mirrors the subcellular localisation of this probe when previously imaged in other in vivo systems (see reference 21, Figure 5A-E; reference 22, Figure S2).

2. Are there any variations among the pupae? In all experiments, the data are summed up and shown by mean and standard error. It would be informative if the difference among the sample is also shown.

To address this point, we have generated dot plots for all FRET graphs. These can be found in Supplementary Figures 1 and 3.

3. Re: Difference in Rac1 activity among the wild-type and mutants: This could be most easily and impressively represented if the XY images of Rac1 activity of chimera can be shown as in Fig. 3C and 3D. Although the data are normalized to the values of apical region of the cell, there should be XY planes that show the difference.

Unfortunately it is not technically possible to show the ratiometric FRET for mutant cells alongside wild type cells within the same image, as is done for Figs 3C and 3D. This is because we require a fluorescence marker to identify the mutant clones, which then makes FRET analysis impossible due to fluorescence interference.

Minor comments:

1. Fig. 2F, G: This may be specific to the reviewer's PC, but the panels are overlain by black sheets.

We do not see this. Hopefully with the Illustrator version of the files you will not have this problem.

Reviewers' comments:

Reviewer #1 (Remarks to the Author):

The authors have addressed most of my comments but I think there are some limitations that should be clarified.

1) Previous comment 2: It is not helpful to show the FRET data normalized as it can lead to misleading interpretations. If I understand the authors correctly, they the normalizations were done for each condition separately; hence, not even the levels of curves within the same graph are comparable. I think such data should be shown in absolute values as it is important to understand the observed phenotypes to know how absolute Rac activities were affected. The observation that the gradient is more shallow in Baz mutants may fit the hypothesis assuming the only way Baz affects Rac is by inhibiting Tiam1. However, this is an extrapolation from other studies and systems and it would go a long way if the absolute values were shown that demonstrate that the gradient is more shallow since Rac activity increases towards the apical end of the lateral membrane in TIAM1 mutants.

2) Previous comments 3&4: It is a such not important whether these protrusions are called microvilli or microvilli-like, the resolution of the images is insufficient to decide whether they changed or not. So, the authors may conclude that there was no overall reduction in apical actin intensity, but the images do not allow statements about possible effects on the structure.

Reviewer #2 (Remarks to the Author):

It has now become clear that the authors modified the prototype Rac1 biosensor to locate the cytoplasm. This is somewhat surprising because Rac is believed to be active at the plasma membrane. Other than this point, the authors answered most, if not all, questions raised by this reviewer.

Response to reviewers.

Reviewer #1:

1) Previous comment 2: It is not helpful to show the FRET data normalized as it can lead to misleading interpretations. If I understand the authors correctly, they the normalizations were done for each condition separately; hence, not even the levels of curves within the same graph are comparable.

We would like to clarify to the reviewer how we normalise our data. For each individual cell that is analysed, FRET is calculated at each z plane from apex to base. Then all FRET values for a given cell are normalised to the FRET value at the apex for that particular cell. So, in essence, we are looking at the difference in FRET (and therefore Rac activity) between the apex of the cell and subsequent z planes basally. Once we have these values, we average all FRET values at each z position for each genotype to construct the graph. We can then compare the average FRET values at each z position between genotypes. This normalisation allows us to directly compare, and to easily visualise, patterns of intracellular differences in Rac activity between different genotypes.

I think such data should be shown in absolute values as it is important to understand the observed phenotypes to know how absolute Rac activities were affected.

We agree with the reviewer that if we were able to use absolute values, it would add valuable information to the study, but unfortunately there are technical limits to the use of this type of probe in a living animal and the use of absolute values would not be informative in this case.

First, the variability in FRET signal between cells is substantially greater than the apicobasal gradient that we observe within individual cells.

When we look at absolute FRET levels within one genotype, e.g. within a population of wild type cells (see graph below), the apicobasal gradient of Rac activity is clearly evident in each cell, however the variability of the FRET signal between cells is relatively large. If we look at the variability among different cells in a single z plane, for example at the apex, the range is approximately 0.9 (from 0.4 to 1.3). This is about 2 fold larger than the actual apicobasal differences within one given cell - the apicobasal gradient that we observe ranges from 0.14 to 0.65, with an average of 0.39 (e.g. dark blue cell in graph below: $1.6 - 1.15 = 0.45$).

Second, we are using the GAL4/UAS system to express the FRET biosensor. The level of transgene expression using the GAL4/UAS system can be very variable, and this is likely to explain the high variability that we see between cells. This variability can be due to a number of factors, including temperature and cell fitness. Fitness amongst cells varies naturally in wild type tissue, but differences in fitness can be greatly increased in mutant genetic backgrounds. It would therefore not be possible to compare absolute values between genotypes, as the level of transgene expression will vary considerably. Additionally, it is likely that the size of the mutant clone, and the position of the transgene expressing cell within a mutant clone, will affect the level of expression. While this variability in expression is likely to affect absolute values, it will not affect the gradient that we see within a given cell. Therefore, normalising to the apical value allows us to compare intracellular gradients between genotypes and allows us to draw meaningful conclusions. We would like to point out that normalising data using in vivo biosensors is common practice, as it usually represents the best way to visualise differences between conditions/genotypes.

The observation that the gradient is more shallow in Baz mutants may fit the hypothesis assuming the only way Baz affects Rac is by inhibiting Tiam1. However, this is an extrapolation from other studies and systems and it would go a long way if the absolute values were shown that demonstrate that the gradient is more shallow since Rac activity increases towards the apical end of the lateral membrane in TIAM1 mutants.

Even though we are unable to use absolute values to resolve this issue, we do consider that we have extensive evidence to suggest that Baz is inhibiting Rac activity to a large degree via Sif/TIAM1 in this system.

1) A *baz* mutant cell phenocopies a Sif overexpressing cell. In both cases, high levels of the Moe-GFP reporter are observed throughout the basolateral domain. Also,

filopodia are no longer restricted to the base of the cell, but are observed all along the lateral membrane, occurring at the expense of the lateral sheet-like protrusions. This same phenotype is also observed when overexpressing a Rac constitutively active transgene (see Georgiou and Baum, 2010).

2) A *sif* mutant cell loses all protrusions in the basolateral domain, phenocopying the overexpression of either bazooka or a Rac dominant-negative transgene (Georgiou and Baum, 2010).

3) Both *baz* and *sif* mutant cells result in a severe disruption to the Rac activity gradient. Indeed, the high degree of correlation that we observe between protrusion form and position and the Rac activity gradient, in all genotypes studied, illustrates the robustness of our data.

2) Previous comments 3&4: It is as such not important whether these protrusions are called microvilli or microvilli-like, the resolution of the images is insufficient to decide whether they changed or not. So, the authors may conclude that there was no overall reduction in apical actin intensity, but the images do not allow statements about possible effects on the structure.

We agree with the reviewer and have modified the text accordingly on pages 5 and 7.

Reviewer #2:

It has now become clear that the authors modified the prototype Rac1 biosensor to locate the cytoplasm. This is somewhat surprising because Rac is believed to be active at the plasma membrane. Other than this point, the authors answered most, if not all, questions raised by this reviewer.

We would just like to point out that we did not modify the Rac Biosensor ourselves, rather it was modified by Kardash et al. (2010). The authors of this paper explained that they modified the biosensor to, "match the subcellular localization of the RacFRET biosensor with that of the EGFP-Rac1 fusion".